# From Low Rank Gradient Subspace Stabilization to Low-Rank Weights: Observations, Theories, and Applications

Ajay Jaiswal [* 1]   Yifan Wang [* 2]   Lu Yin [3 4]   Shiwei Liu [3 5]   Runjin Chen [1]   Jiawei Zhao [6]   Ananth Grama [2]
Yuandong Tian [7]   Zhangyang Wang [1]

## Abstract

Large Language Models' (LLMs) weight matrices can often be expressed in low-rank form with potential to relax memory and compute resource requirements. Unlike prior efforts that focus on developing novel matrix decompositions, in this work we study the non-uniform low-rank properties of weight matrices in LLMs through the lens of stabilizing gradient subspace. *First,* we provide a theoretical framework to understand the stabilization of gradient subspaces through Hessian analysis. *Second,* we empirically establish an important relationship between gradient dynamics and low-rank expressiveness of weight matrices. Our findings reveal that different LLM components exhibit varying levels of converged low-rank structures, necessitating variable rank reduction across them to minimize drop in performance due to compression. Drawing on this result, we present *Weight Low-Rank Projection* **(WeLore)** that unifies weight compression and memory-efficient fine-tuning into one, in a data-agnostic and one-shot manner. When used as a compression technique, WeLore categorizes weight matrices into Low-rank Components (LRCs) and Non-Low-rank Components (N-LRCs) and suitably encodes them for minimum performance loss. Our gradient dynamics perspective illustrates that *LRCs tend to have better finetuning capabilities* and their standalone finetuning can closely mimic and sometimes outperform the training loss trajectory and performance of full-finetuning with notable memory and compute footprint reduction. Codes are available at https://github.com/VITA-Group/WeLore.

---
[*]Equal contribution  [1]University of Texas at Austin [2]Purdue University [3]Eindhoven University of Technology [4]University of Surrey [5]University of Oxford [6]California Institute of Technology [7]Meta AI. Correspondence to: Ajay Jaiswal <ajayjaiswal@utexas.edu>.

*Proceedings of the $42^{nd}$ International Conference on Machine Learning*, Vancouver, Canada. PMLR 267, 2025. Copyright 2025 by the author(s).

## 1. Introduction

Large language models (LLMs) are designed with a large number of parameters to capture complex patterns and relationships in data. However, despite their huge size with billions of parameters, these models' weights often exhibit low-rank structures in practice, offering potential to relax memory and compute resource requirements. While many efforts have focused on developing novel matrix decompositions for efficient training (Hu et al., 2021b; Dettmers et al., 2023; Meng et al., 2024; Biderman et al., 2024b; Lialin et al., 2023a) and model compression (Hsu et al., 2022b; Kaushal et al., 2023b; Li et al., 2023; Jaiswal et al., 2024; Wang et al., 2023b), the underlying mechanisms of how these low-rank structures emerge remain insufficiently understood.

A fundamental question arises: *Why do the weights of LLMs tend to become low-rank?* Prior efforts have proposed various explanations – one perspective focuses on the inherent distribution of training data, which may reside on a low-dimensional manifold (Timor et al., 2022; Ergen & Pilanci, 2021; Ongie & Willett, 2022). For instance, when data lie on a $d$-dimensional manifold, the weight matrices of a two-layer ReLU network converge to a rank of at most $d$ (Ongie & Willett, 2022). Another explanation examines the optimization dynamics, where certain training procedures (*e.g.*, SGD with weight decay) naturally favor convergence to low-rank regions (Ji & Telgarsky, 2020; Le & Jegelka, 2022; Galanti et al., 2024).

In this work, we approach the low-rank phenomenon through the lens of gradient subspace stabilization. Gradient dynamics during pretraining plays a crucial role in shaping the weight subspace, acting like a stream that gradually sculpts the weight landscape. Through theoretical analysis of gradient subspace stabilization via Hessian analysis, we establish a fundamental connection between gradient dynamics and the emergence of low-rank structures in LLMs. A key insight from our analysis is the presence of a distinct Hessian gap, indicating the emergence of a low-rank subspace in the weight space. Through accompanying experiments, we identify two significant findings:

1. Different components of LLMs exhibit varying degrees

of low-rank convergence: ① MLP Up/Down Projections and Self-Attention Value Projections demonstrate an unclear Hessian gap and slow gradient subspace convergence, resulting in high-rank weights. ② Self-Attention Query/Key and MLP Gate Projections display clear Hessian gaps, and gradient subspaces settle rapidly, resulting in low-rank weights.

2. The low-rank convergence patterns vary across different network depths: ① Middle layers exhibit small Hessian gaps and slow gradient subspace convergence, resulting in high-rank weights. ② Early and late layers have clear Hessian gaps and rapidly evolve into low-rank subspaces, resulting in low-rank weights.

Based on these observations, we categorize weight matrices into two groups: ① **Low-rank Components (LRCs)**: Matrices exhibiting high-quality low-rank structure (characterized by heavy-tail in sorted singular values) whose gradients carry rich error signals from data. ② **Non-Low-rank Components (N-LRCs)**: Matrices with non-converged low-rank structure (missing heavy-tail in distribution of singular values) that cannot be effectively low-rank factorized.

These conclusions lead to our proposed Weight Low-Rank Projection (**WeLore**) approach, which unifies weight compression and parameter-efficient fine-tuning as one:

1. **Compression perspective (WeLore-COMP) :** LRCs with stabilized low-rank structure can be factorized by SVD to achieve significant compression ratios.

2. **Parameter-Efficient Finetuining perspective (WeLore-PEFT) :** During fine-tuning, we selectively update only LRCs in their low-rank decomposed format while keeping N-LRCs frozen, leading to effective gradient progress with reduced memory footprint.

WeLore operates in a one-shot, data-agnostic manner, performing layer-wise non-uniform rank reduction based on the heavy-tail property of normalized singular values. This approach allows LRCs to support higher rank reduction while preserving N-LRCs at full rank or minimal reduction, ensuring minimal performance degradation. Notably, WeLore achieves remarkable efficiency gains: compared to full-finetuning of a 50% compressed LLaMA-2 7B model, WeLore requires only 35% of trainable parameters, delivers 3× better throughput, and reduces GPU memory usage by 40%. Most importantly, our extensive experiments across both continual finetuning (Figure 5) and downstream tasks (Figure 6) demonstrate that this selective LRC-based fine-tuning can match or even outperform full model finetuning, establishing WeLore as a powerful unified solution for both model compression and efficient adaptation.

## 2. Gradient-Hessian Eigenspace Alignment to Low Rank Weights: Theory and Observations

In this section, we present a theoretical framework for understanding the stabilization of gradient subspaces through Hessian analysis. We provide insights into the convergence properties of Hessian eigenspaces, showing that the gradient subspace aligns with the dominant directions in the Hessian. This analysis is crucial for understanding the emergence of low-rank structures in LLMs.

### 2.1. Preliminaries

We denote the training loss as $L(W)$, where $W \in \mathbb{R}^{m \times n}$ is the set of model parameters. Its gradient is denoted by $\nabla L(W) \in \mathbb{R}^{m \times n}$, and the Hessian, as a linear operator, is denoted by $H(W) = \nabla^2 L(W) \in \mathbb{R}^{mn \times mn}$.

We adopt the standard SGD update

$$W_{t+1} = W_t - \eta \nabla \mathcal{L}(W_t),$$

with learning rate $\eta > 0$.

The following analysis relies on several assumptions, which we summarize here and present in detail in Appendix A.1). We assume the Hessian is Lipschitz continuous, which is justified by the smoothness of common loss functions like cross-entropy. The loss function satisfies the Kurdyka–Łojasiewicz (KŁ) condition near optimum, generalizing convexity to the nonconvex setting. We also require a uniform spectral gap between dominant and non-dominant eigenvalues of the Hessian, which naturally emerges from the dominance of task-relevant directions in neural networks. Finally, we assume the model architecture is reversible (Tian et al., 2021), ensuring bounded norms.

### 2.2. Hessian Eigenspace Stabilization

Our first important theoretical result shows that both the eigenvalues and eigenspaces of the Hessian converge during training.

**Theorem 2.1.** *Let $H_t = \nabla^2 L(W_t)$ be the Hessian of the loss $L(W)$ at time $t$. Under standard assumptions, the eigenvalues and eigenspaces of $H_t$ stabilize as $t \to \infty$. Specifically:*

①*The eigenvalues converge with step-to-step changes decaying to zero;*

②*The top-$r$ eigenspace converges with vanishing step-to-step changes.*

*Proof Sketch.* The proof follows three key steps. First, we bound the step-to-step changes in the Hessian using Lipschitz continuity: $\|\Delta H_t\| \leq L_H \|W_{t+1} - W_t\| =$

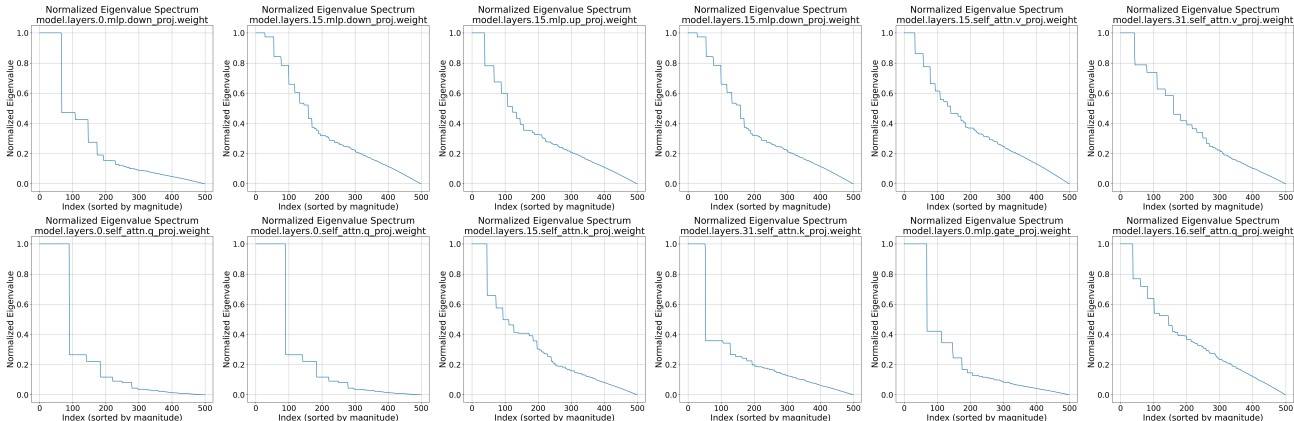

*Figure 1.* Hessian gap across layers and components of LLaMA2-7B. We observe that: 1) mlp.up_proj, mlp.down_proj, and self_attn.v_proj exhibit less pronounced Hessian gaps compared to self_attn.k_proj, self_attn.q_proj, self_attn.o_proj, and mlp.gate_proj; 2) early and late layers generally display clearer gaps than middle layers; 3) components with a pronounced Hessian gap (self_attn.k_proj, self_attn.q_proj, self_attn.o_proj, and mlp.gate_proj) tend to be more low-rank, as shown in our experiments.

$\eta L_H \|\nabla L(W_t)\|$. Second, we leverage the KŁ condition to show that the gradient norm decay follows a power law: $\|\nabla L(W_t)\| \leq C/t^{\theta/(1-\theta)}$. This gives us $\|\Delta H_t\| \leq \eta L_H C/t^{\theta/(1-\theta)}$. Finally, we use Weyl's inequality to bound eigenvalue changes and the Davis-Kahan theorem to control eigenspace shifts, showing both converge as series $\sum 1/t^{\theta/(1-\theta)}$ for $\theta > \frac{1}{2}$. Full proof in Appendix A.2. □

### 2.3. Gradient-Hessian Eigenspace Alignment

Building on the stability results, we show that gradients naturally align with the principal eigenspaces of the Hessian, explaining why gradients eventually operate in a low-dimensional subspace.

**Theorem 2.2.** *Under the conditions above, the gradient vector $G_t = \nabla_W L(W_t)$ asymptotically aligns with the principal eigenspace of the Hessian. Specifically:*

$$\lim_{t \to \infty} \frac{\left\| \left( I - U_t U_t^\top \right) G_t \right\|}{\|G_t\|} = 0$$

*where $U_t$ spans the dominant eigenspace of $H_t$.*

*Proof Sketch.* The proof analyzes how gradients evolve in the eigenspace basis of the Hessian. First, we decompose the gradient update using Taylor expansion: $G_{t+1} = G_t - \eta H_t G_t + O(\eta^2 \|G_t\|^2)$. Next, we project this onto the non-dominant subspace $(I - U_t U_t^\top)$ and use the spectral gap condition $\lambda_j \leq \lambda_r - \gamma$ for $j > r$ to show the non-dominant component contracts: $\|(I - U_t U_t^\top) G_{t+1}\| \leq (1 - \eta\gamma/2)\|(I - U_t U_t^\top) G_t\|$. This geometric contraction, combined with bounded $\|G_t\|$ from reversibility, proves that the gradient asymptotically aligns with the dominant eigenspace. Full details are presented in Appendix A.3. □

### 2.4. Hessian Eigenspace Dynamics and Gradient Subspace Stablization

The above analysis gives us a key insight connecting gradient subspace stabilization with Hessian properties. A clear **spectral gap**, defined as the difference between the dominant and non-dominant eigenvalues of $H_t$, facilitates rapid gradient alignment with principal subspaces. **Components with a clear spectral gap in their Hessian eigenvalues stabilize rapidly, while layers with a flatter spectrum exhibit slower or no stabilization.**

In Figure 1, we observe the second-order derivative of the loss $L(W)$ with respect to model parameters $W$ for each layer's component. Components such as query, key, and gate projections benefit from this property, as their roles inherently induce low-rank structure in the Hessian. In contrast, value and down projections lack this gap due to diffuse gradient contributions, compression effects, and weaker signals, resulting in flatter spectra. Additionally, we find that early and late layers tend to develop clear Hessian gaps, whereas middle layers typically show a flatter spectrum.

The following discussion illustrates how a component's role and position in the Transformer architecture affect Hessian stablization:

**Self-Attention Q/K Projections** These layers exhibit a low-rank structure, as softmax reweighting in attention : only a few tokens contribute significantly to the attention scores. These strong Hessian spectral gaps allow gradients to align with principal subspaces and stabilize rapidly.

**Self-Attention V Projection** By contrast, the value projection processes token content after attention scores are determined, yielding a denser Hessian without a clear spectral gap. This diffuse spectrum hinders gradient alignment

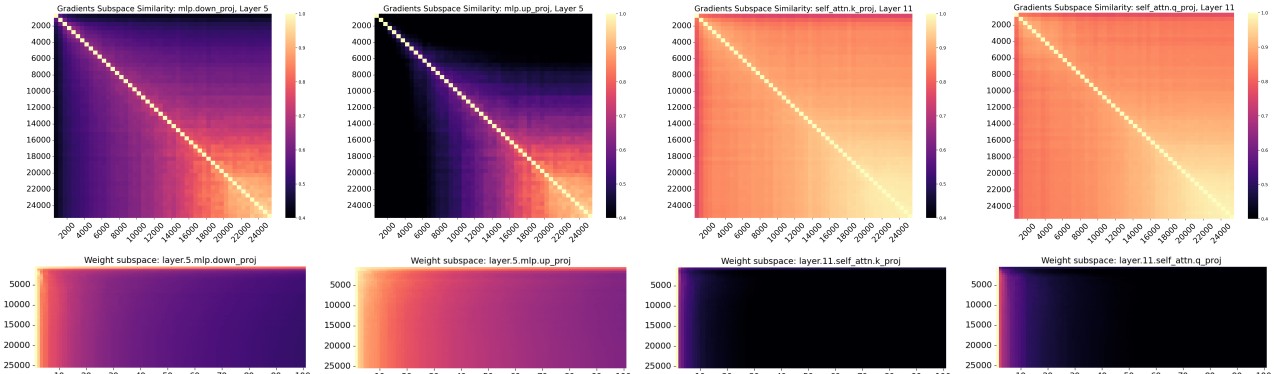

*Figure 2.* (Row 1) Gradients subspace similarity obtained from various checkpoints during pretraining of LLaMA-130M on C4 dataset for 25,000 training steps using Adam Optimizer. (Row 2) Emergence of Low-rank Weight Subspace during pretraining of LLaMA-130M. Each row of individual subplot represents the singular values of weights in a given training step.

and delays stabilization.

**MLP Gate Projections**    Activations like GELU selectively amplify dominant features, creating sparse gradients that concentrate in a few principal directions. Consequently, the Hessian is low-rank with a strong spectral gap, enabling swift gradient convergence.

**MLP Up and Down Projections**    The down projection compresses high-dimensional features back to the model's base dimension, flattening the Hessian spectrum and diluting the strong-gradient directions. This lack of a clear spectral gap slows or prevents stabilization.

**Layer-Wise Trends**    Early layers stabilize rapidly due to simpler input representations and the low-rank structure of embeddings and positional encodings. Late layers also stabilize effectively, as gradients become dominated by task-relevant features propagated from the loss function. In contrast, middle layers exhibit delayed stabilization, as they aggregate and mix features from multiple layers, introducing gradient diffusion. Additionally, middle layers are more affected by attenuation from softmax and LayerNorm, which further delays stabilization.

### 2.5. Weight Low-rank Dynamics in Pre-trained LLMs

Our aforementioned spectral gap analysis of Hessian eigenvalues illustrates the non-uniform stabilization of gradient subspace. Provided the pretrained LLMs weights are essentially accumulation of gradients over the course of pretraining, an important question to ask is: *How do the low-rank properties of LLM weights correlate with their corresponding gradient dynamics stabilization?* In this section, we study how gradient subspace stablization observations translate to the emergence of low-rank structures in the weight matrices of the model. Figure 2 (row 2) presents the corre-

sponding emergence of weight low-rank structures throughout pretraining within different layers of LLaMa-130M using C4 daatset. Our findings are summarized as:

- We find the emergence of low-rank structure across the weight matrices very early during pretraining, which becomes explicit and notable as pretraining progresses.

- Similar to our gradient subspace observations, we find that not all layers can express themselves as low-rank and this property significantly varies subject to position (middle layers or terminal layers) and role (attention layers or MLP layers).

- We empirically find a strong correlation between the gradient subspace stabilization and the low-rank emergence across the weight matrices (*e.g.,* the absence of a clear spectral gap that prevents stabilization within `model.layer.5.mlp.down_proj` reflects in the weight matrix not converging to low-rank[1])

Next, we investigate whether our low-rank findings are valid for the weight matrices within a publicly available LLM checkpoint. Figure 3 presents the 4096 normalized singular values corresponding to different layers across 32 transformer blocks of LLaMa-2 7B. It can be clearly observed that some layers (*e.g.,* `self_attn.q_proj`, `self_attn.k_proj`) elicit a heavy tail behaviour indicating better low-rank expressivity compared to others (*e.g.,* `mlp.up_proj`, `mlp.down_proj`). Another important observation of note is that majority of the layers from the front and tail blocks of the model tend to have better low-rank property, which is consistent with our gradient behavior

---

[1]A sharp bright line across the subplots in Figure 2 (row 2) to the left suggests heavy-tail distribution of singular values. A heavy-tail singular value distribution is a favorable property that indicates the matrix can be well compressed using a few singular vectors without introducing large reconstruction errors.

analysis. Heavy tail indicates that only a small fraction of singular values carries maximum information and the corresponding matrix can be well approximated using a fraction of basis vectors from SVD with small reconstruction error.

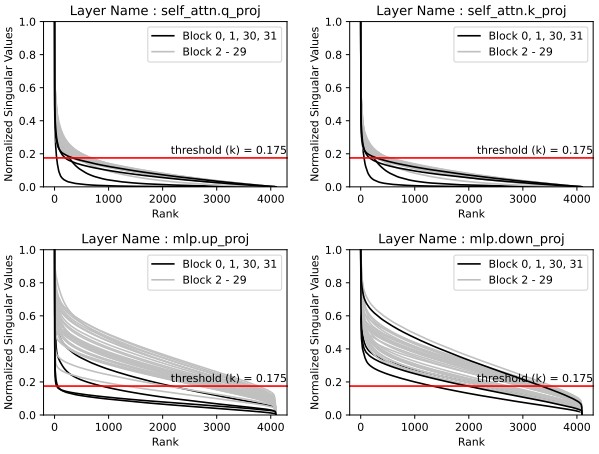

*Figure 3.* Normalized singular values of the weight matrices corresponding to different layers of LLaMa-2 7B pretrained checkpoint from HuggingFace. Each subplot indicates 4096 sorted and normalized singular values corresponding to a layer (*e.g.,* `self_attn.q_proj`) from 32 transformer blocks.

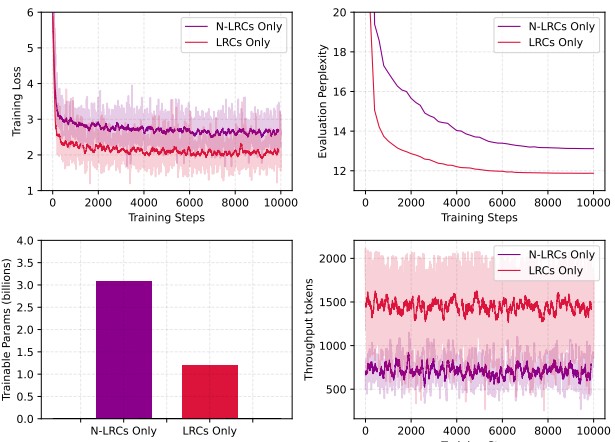

*Figure 4.* Finetuning statistics and performance comparison of Low Rank Components (LRCs) and Non-Low-Rank Components (N-LRCs) layers of a 50% compressed LLaMa-2 7B model with C4. Note that all finetuning hyperparameters are kept same in both settings for fair comparison.

## 3. Adaptive Low-rank Weight Projections: Applications

### 3.1. WeLore: Low-rank Compression Technique

Among multiple techniques for LLM compression, low-rank decomposition of pretrained weights as a product of two smaller dense matrices receives special attention because it can leverage highly optimized floating-point dense matrix

multiplication kernels. This is unlike sparsity and quantization, which require specialized kernels, often optimized for each hardware platform for best performance. Recently, several efforts (Hsu et al., 2022a; Kaushal et al., 2023a; Yuan et al., 2023; Wang et al., 2023a; Saha et al., 2023; Wang et al., 2024) have explored matrix factorization of LLMs' pretrained weights. These efforts primarily focus on improving SVD using more informative signals like activation, Fisher information and applying it uniformly (same rank reduction ratio) across all the weights. As discussed in previous section, low-rank emergence varies significantly across candidate weights in a pretrained checkpoint. To this end, we pose a relatively under-explored question: *How can we carefully curate a layer-adaptive rank reduction ratios for all layers in the pretrained checkpoint?*

**WeLore-COMP** (WeLore for compression) is a data-agnostic and implementation-friendly normalized singular value thresholding technique[2] with only one global hyper-parameter (threshold k), as shown as the shaded red and green region in Figure 3 for layer-adaptive rank reduction. Specifically, we preserve normalized singular values greater than the threshold k, shown as shaded green region. For a given effective rank reduction ratio[3] of $ERR$, the global threshold k can be approximated using linear search[4] over `np.linspace(0, 1, 0.005)` with condition as follows:

$$\frac{\sum_l \texttt{sum}(\mathcal{S}_{W_l} < \texttt{k})}{\sum_l \texttt{len}(\mathcal{S}_{W_l})} \approx ERR \qquad (1)$$

Here, $W_l$ represents the weight matrix of layer $l$ and $\mathcal{S}_{W_l}$ is the array of sorted normalized singular values estimated with `torch.svd`$(W_l)$. Note that k estimation is not computationally expensive, since $\mathcal{S}_{W_l} \forall l$ can be calculated before searching for k. Given a weight matrix $W_l^{4096 \times 4096}$ and $\mathcal{S}_{W_l} = \{s_1, s_2, ..., s_{4096}\}$, the compressed rank $r$ can be computed as $r = \texttt{np.sum}(\mathcal{S}_{W_l} \geq \texttt{k})$. In compressed format, $W_l^{4096 \times 4096}$ can be represented as a composition of two small matrices $A_l^{4096 \times r}$ and $B_l^{r \times 4096}$ where $r << 4096$. As observed in Figure 3, for k = 0.175, which indicates an aggregated 50% rank reduction, majority of the `self_attn.q_proj` from 32 transformer blocks of LLaMa-7B can undergo significant reduction $\geq$ 90% (*i.e.,* $r < 400$). On the other hand, layers such as `mlp.up_proj` & `mlp.down_proj`, which are not low-rank friendly, receive high $r$.

Given $r_l$ for all layers $l$ in the pretrained checkpoint, WeLore categorizes all the layers into two broad categories - Low-rank Components (LRCs) and Non-Low-rank Components

---

[2]Normalization helps us to compare singular value distribution across all layers at the same scale.

[3]Effective Rank Reduction (ERR): $1 - \frac{\sum_l rank(W_l^{Compressed})}{\sum_l rank(W_l^{Original})}$

[4]Pseudo-code for k estimation is provided in Appendix D.

| | LLaMa2-7B [PPL: 7.03] | | | LLaMa2-13B [PPL: 6.53] | | | Mistral-7B [PPL: 8.14] | | |
|---|---|---|---|---|---|---|---|---|---|
| Rank Reduction | Uniform Reduction | OWL Reduction | WeLore-COMP Reduction | Uniform Reduction | OWL Reduction | WeLore-COMP Reduction | Uniform Reduction | OWL Reduction | WeLore-COMP Reduction |
| 10% | 10.58 | 12.11 | 7.13 | 7.17 | 7.2 | 6.55 | 12.31 | 11.63 | 8.76 |
| 20% | 16.43 | 14.49 | 8.28 | 8.61 | 8.53 | 6.96 | 78.69 | NaN | 11.90 |
| 30% | 91.99 | NaN | 14.41 | 13.99 | 11.63 | 8.66 | 6746.48 | NaN | 30.69 |
| 40% | NaN | NaN | 78.17 | 1178.03 | 56.06 | 24.92 | 162301.04 | NaN | 429.08 |
| 50% | NaN | NaN | 1836.62 | 4167.79 | 7984.39 | 1142.53 | 248042.97 | NaN | 1351.32 |

*Table 1.* Perplexity (C4 dataset) comparison of LLaMa-2 7B, 13B and Mistral-7B pretrained checkpoints compressed with WeLore-COMP, Uniform, and outlier-sensitive rank reduction ratios.

(N-LRCs). Layers with heavy-tail which can be effectively represented with $r_l < 0.5 \times \text{rank}(W_l)$ fall in LRCs, while the rest fall in N-LRCs. We replace weight matrices of all LRCs in pretrained checkpoint by composition of two small matrices $A$ & $B$ to achieve notable parameter reduction (*e.g.*, $\times 0.67$ parameters with $\mathbf{R} = 0.5$) saving memory and compute during inference and fine-tuning (low-rank weight representation allows gradients and optimizer states to be in low-rank during finetuning).

### 3.2. WeLore: Parameter-Efficient Finetuning Technique

Parameter-Efficient finetuning techniques (PEFT) which enable LLMs to perform a new task with minimal updates has received enormous attention to their ability to allow finetuned by only updating a small number parameters. Unlike LoRA and its varients which finetune a *small added fraction* of parameters to original pretrained weight checkpoints not relevant to original pretraining optimization, WeLore provides an alternative approach by capitalizing the gradient perspective to *select a small fraction of weights* from the pretrained model which can undergo fine-tuning. As discussed above, LRCs exhibits low-rank structure with rich gradient dynamics while N-LRCs can't be well-expressed in low-rank format. To this end, **WeLore-PEFT** make the following proposal:

Given a low-rank compressed checkpoint with LRCs and N-LRCs, finetuning with backpropagation **only through LRCs** (frozen N-LRCs) can closely mimic the performance of full-finetuning (sometimes better) with considerable memory and compute reduction. Note that LRCs are represented in low-rank format, both gradients and optimizer state will by default in low-rank saving finetuning cost.

**Empirical evidence that LRCs are better at learning than N-LRCs:** Here, we investigate the relative difference in performance and quantify compute cost related to finetuning LLMs. Figure 4 presents a comparison of continual finetuning statistics of LLaMa-7B pretrained checkpoint with 50% effective rank reduction ratio on C4 dataset for 10,000 training steps. Red color indicates finetuning by back-propagating only through LRCs (freezing all the N-LRCs) while magenta color indicates finetuning N-LRCs (freezing LRCs). It can be clearly observed that despite

$\sim 3\times$ more trainable parameters, training loss as well as the validation perplexity of finetuning N-LRCs are significantly *under-performing* in comparison to finetuning LRCs. Moreover, it is important to note that the throughput achieved by LRCs is $\sim 2\times$ in comparison to N-LRCs, which can be attributed to the parameter-efficient low-rank represented weight matrices, gradients, and optimizer state.

## 4. Experiments and Analysis

In this section, we first estblish the superiority of WeLore's layer-adaptive rank reduction ratio for effective low-rank compression of pre-trained checkpoints of LLMs. Next, we demonstrate the effectiveness of WeLore for LRCs-focused parameter efficient finetuning performance across several downstream tasks. Unlike prior efforts that *either focus on low-rank compression or parameter-efficient finetuning*, WeLore **uniquely** differentiates itself by proposing an effective low-rank compression strategy and presents a novel angle of memory and parameter-efficient fine-tuning using LRCs while yielding performance comparable to full-finetuning.

### 4.1. WeLore for Compression of Pre-trained LLMs

We evaluate the overall performance of the WeLore-COMP non-uniform rank reduction ratio from three aspects: ① perplexity-based performance evaluation in comparison with uniform rank reduction and outlier-sensitive rank reduction; ② performance evaluation on real-world tasks (*e.g.,* summarization, factoid-QA, and multi-turn conversation), ③ performance comparison of WeLore-COMP when combined with activation-guided SVD unlike conventional SVD of LLM weight matrices. Additional empirical GPU memory requirement statistics for varying compression ratios are provided in Appendix E to highlight WeLore usefulness.

*Firstly,* low-rank decomposition of LLMs has been primarily investigated with unilateral (same rank) reduction across all the weights. In contrast, WeLore-COMP presents non-uniform rank reduction ratio guided by emerged low-rank structures in pretrained checkpoints. Table 1 presents the comparison of perplexity of LLaMa-2 7B, 13B and Mistral-7B model checkpoints on C4 validation dataset with EER

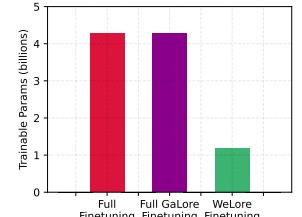
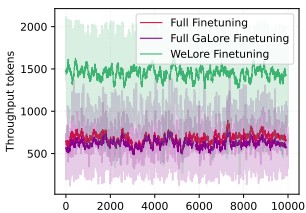

*Figure 5.* Continual-Finetuning statistics and performance comparison of a 50% low-rank compressed LLaMa-2 7B pretrained checkpoint. With exactly same hyperparamter configrations, *WeLore-PEFT can **outperform** full-finetuning* with merely ∼**35%** of trainable parameters, while providing ∼**3× better throughput**.

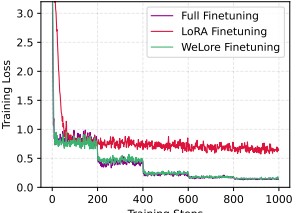
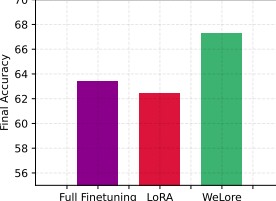
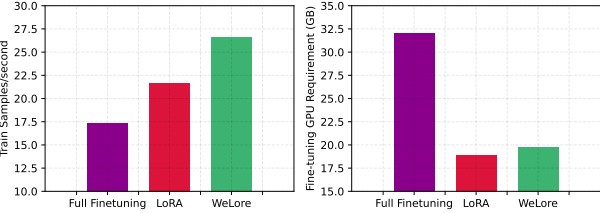

*Figure 6.* Downstream Finetuning statistics and performance comparison of WeLore vs. full-finetuning and LoRA of a 50% compressed LLaMa-2 7B model with StrategyQA dataset with `max_len` of 512. All finetuning hyperparameters are kept identical for fair comparison.

| Method (25% Compression) | Factoid-based QA | Multi-turn Conversation | In-context Summarization |
|---|---|---|---|
| Full Model | 79.02 | 7.61 | 8.15 |
| Uniform Reduction | 34.63 | 4.88 | 5.01 |
| OWL Reduction | 54.42 | 5.27 | 5.16 |
| SVD-LLM | **71.95** | 5.79 | 6.12 |
| WeLore-COMP Reduction | 71.89 | **6.09** | **6.46** |

*Table 2.* Performance comparison of LLaMa-7B compressed (25% ERR) using WeLore-COMP non-uniform rank reduction ratio on real-world benchmarking tasks.

| Method | 10% | 20% | 30% | 40% | 50% |
|---|---|---|---|---|---|
| SVD | 10.58 | 16.43 | 91.99 | NaN | NaN |
| ASVD | 7.24 | 7.75 | 8.85 | 11.33 | 17.03 |
| SVD-LLM | 7.45 | 7.58 | 7.98 | 10.04 | 16.61 |
| WeLore-COMP + ASVD | 7.05 | 7.21 | 7.87 | 9.75 | 14.76 |

*Table 3.* Perplexity (C4 dataset) comparison of LLaMa-7B with SoTA compression techniques at different reduction ratios.

| Model | Method | 10% | 20% | 30% | 40% | 50% |
|---|---|---|---|---|---|---|
| LLaMa-7B | No Finetune | 7.13 | 8.28 | 14.41 | 78.17 | 1836.62 |
| | WeLore-PEFT | 7.15 | 7.40 | 8.18 | 9.47 | 11.87 |
| LLaMa-13B | No Finetune | 6.55 | 6.96 | 8.66 | 24.92 | 1142.53 |
| | WeLore-PEFT | 6.55 | 6.68 | 7.42 | 8.69 | 11.40 |
| Mistral-7B | No Finetune | 8.76 | 11.90 | 30.69 | 429.08 | 1351.32 |
| | WeLore-PEFT | 8.32 | 8.92 | 9.71 | 14.85 | 21.37 |

*Table 4.* Perplexity (C4 dataset) comparison of LLaMa-7B with SoTA compression techniques at different reduction ratios.

| | LLaMa2-7B [1×] | | | | LLaMa2-13B [1×] | | | |
|---|---|---|---|---|---|---|---|---|
| WeLore-COMP → | 30% | 50% | 60% | 70% | 30% | 50% | 60% | 70% |
| Compressed Params | 0.85× | 0.67× | 0.56× | 0.45× | 0.83× | 0.64× | 0.53× | 0.43× |
| + LoRA Finetuning | 8.21 | 12.48 | 21.23 | 382.24 | 7.49 | 21.53 | 27.99 | 124.44 |
| + Galore Finetuning | 9.02 | 18.57 | 396.05 | 670.29 | 8.02 | 60.07 | 2454.03 | 3396.19 |
| + WeLore-PEFT | 8.18 | 11.87 | 17.87 | 47.92 | 7.42 | 11.40 | 19.20 | 73.59 |

*Table 5.* Performance (perplexity) comparison of compressed LLaMa-2 7B & 13B with WeLore-COMP and continual finetuning with LoRA and GaLore w.r.t. WeLore.

of 10% to 50%. It can be clearly observed that as EER increases, the perplexity of the baseline compressed model significantly explodes, but WeLore-COMP retains the perplexity within a reasonable range. For example, WeLore-COMP is ∼ 6.4 × better than 30% Uniform EER for LLaMa-2 7B and ∼ 47 × better than 40% Uniform EER for LLaMa-2 13B. Note that OWL reduction tends to perform sometimes better than Uniform reduction, but its degradation in performance with increasing EER is more severe.

*Secondly,* to further understand the effectivenss of WeLore-COMP in real-world task settings, we consider three popular tasks (factoid-QA, in-context summarization, and multi-turn conversation) following the settings described in Appendix D. For open-ended conversation and in-context summarization tasks, WeLore-COMP significantly outperforms all

baselines. However, it slightly underperforms SVD-LLM for factoid-based question answering task but it is imporatnt to note that SVD-LLM relies on caliberation dataset. Our further studies indicate that WeLore-COMP can be achieve significantly higher performance when it is augmented with advanced activation-guided SVD techniques.

*Thirdly,* Activation-guided SVD techniques (Yuan et al., 2023; Wang et al., 2024) have been found more effective than weight-oriented SVD methods by managing activation outliers and adjusting the weight matrix based on the activation distribution. Despite our work focusing on simple weight SVD to enable easy adaptation and minimize sensitivity to calibration datasets, we conducted experiments to illustrate that WeLore-COMP can also significantly ben-

| | Method | CommonsenseQA | SVAMP | BoolQ | CoinFlip | BigBench[5] | StrategyQA |
|---|---|---|---|---|---|---|---|
| | Dense Full Finetune | 77.052 | 40.672 | 88.189 | 75.000 | 83.742 | 69.581 |
| | Dense LoRA Finetune | 76.414 | 50.090 | 70.962 | 69.333 | 80.995 | 68.690 |
| | Dense GaLore Finetune | 75.339 | 41.667 | 68.362 | 65.667 | 77.980 | 67.325 |
| | + Full Finetune | 75.925 | 40.667 | 84.005 | 51.333 | 83.364 | 70.783 |
| 30% | + LoRA | 64.537 | 44.333 | 81.776 | 61.333 | 68.750 | 65.255 |
| WeLore-COMP | + GaLore | 64.015 | 42.667 | 80.892 | 55.333 | 75.735 | 62.490 |
| | + WeLore-PEFT | 76.744 | 53.333 | 85.040 | 98.667 | 81.818 | 69.648 |
| | + Full Finetuning | 71.908 | 38.333 | 83.603 | 49.000 | 90.224 | 68.502 |
| 40% | + LoRA | 54.386 | 36.667 | 75.021 | 54.667 | 76.002 | 65.154 |
| WeLore-COMP | + GaLore | 52.078 | 36.333 | 71.039 | 50.333 | 77.910 | 65.440 |
| | + WeLore-PEFT | 76.003 | 42.667 | 81.646 | 98.666 | 87.857 | 67.794 |
| | + Full Finetuning | 70.120 | 25.333 | 80.113 | 53.333 | 89.431 | 63.411 |
| 50% | + LoRA | 35.382 | 23.667 | 75.482 | 50.667 | 54.022 | 62.408 |
| WeLore-COMP | + GaLore | 35.122 | 21.667 | 71.552 | 47.667 | 58.975 | 61.336 |
| | + WeLore-PEFT | 70.516 | 30.667 | 80.377 | 94.666 | 87.802 | 67.290 |

*Table 6.* Downstream performance of Dense and WeLore compressed LLaMa-2 7B checkpoint under full-finetuning along with memory-efficient finetuning techniques (LoRA and GaLore). All downstream finetuning is performed starting from the same initial checkpoint state for fair comparison.

efit from Activation-SVD. Table 3 presents the perplexity comparison of SoTA low-rank compression techniques (including SVD-LLM and ASVD which rely on calibration datasets) wrt. WeLore-COMP non-uniform layer-wise ratios when augmented with calibration dataset.

## 4.2. WeLore for Memory-efficient Finetuning of Pre-trained LLMs

In this section, we investigate the effectiveness of WeLore-PEFT, which focuses on selective finetuning of LRCs components to answer three important questions:

① *How well can WeLore-PEFT recover the language modeling capabilities of compressed LLMs?* We investigate the performance characteristics of LRC-focused WeLore-PEFT in recovering compressed pretrained checkpoints perplexity of LLaMa-2 7B, 13B and Mistral 7B. More specifically, given a pretrained checkpoint, we first perform rank reduction using WeLore-COMP with varying ERR between 10-50%. Table 4 presents the C4 (evaluation set) perplexity of SVD of weight matrices. Note that with increasing ERR, model perplexity is notably high. Next, we perform limited continual finetuning of a fraction of LRCs using C4 dataset with sequence length of 1024 on 0.7M tokens. Interestingly, it can be observed from Table 4, that WeLore-PEFT can significantly recover the perplexity of compressed checkpoints within a reasonable range, where benefits increase with higher compression ratios.

② *How does WeLore-PEFT compare with conventional LLM finetuning techniques like LoRA and GaLore?* To investigate the effectiveness of WeLore-PEFT in comparison with two popular resource-efficient finetuning methods (LoRA and GaLore), we designed controlled experiments with fixed

training token budget (0.7 million) from C4 dataset and keeping all other hyperparameters indetical for fair comparison. Table 5 illustrates the superiority of LRCs-focused WeLore-PEFT, which achieves $\sim 8\times$ and $\sim 1.7\times$ at 70% compression ratio for LLaMa-7B and 13B models.

③ *How well WeLore-PEFT performs for finetuning on benchmarking downstream tasks?* To understand the effectiveness of LRCs-only WeLore-PEFT finetuning, we consider full-parameter finetuning, LoRA, and GaLore for dense pretrained checkpoint, as well as WeLore compressed checkpoint of LLaMa-7B. We conduct several experiments across various compression ratios on math and commonsense reasoning tasks and report our performance in Table 6. Surprisingly, LRCs-based finetuning of compressed models tends to closely match and sometime outperform even the dense, as well as compressed full-parameter finetuning of LLaMa-7B pretrained checkpoint. Additionally, the performance achieved by WeLore-PEFT is significantly and consistently higher than both LoRA and GaLore across all the tasks with memory requirements close to LoRA. Figure 6 shows that unlike LoRA, WeLore-PEFT closely mimics the loss trajectory of full-finetuning with significantly lower GPU memory requirements and can achieve throughput greater than LoRA based fine-tuning.

## 5. Conclusion

In this work, we focus on the emerging non-uniform low-rank properties across weight matrices in LLMs through the lens of stabilizing gradient subspace. We provide a theoretical framework to understand the stabilization of gradient subspaces through Hessian analysis. We empirically demonstrate a consequential relationship between gradient

dynamics and low-rank expressiveness of weight matrices. We present WeLore-COMP, an adaptive layer-wise low-rank compression strategy for low-rank decomposition, which achieves high compression with minimal drop in performance. The unique proposition of WeLore lies in categorizing weight matrices of pretrained models into two broad categories - LRCs and N-LRCs based on their low-rank structure. We conduct extensive experiments to validate that LRCs yield better trainability than N-LRCs. Given limited compute & memory budget, WeLore-PEFT proposes finetuning of LRCs while keeping N-LRCs frozen with back-propagation for maximal gain (sometimes better than full-finetuning).

## Impact Statement

This paper presents a theoretical framework to understand the stabilization of gradient subspaces through Hessian analysis. It proposes WeLore, a joint compression and efficient-finetuning strategy grounded on our findings of the existence of varying levels of converged low- rank structures across different LLM components. Our work could lead to improved methods for efficient LLMs development, and contribute to the "GreenAI" goal.

## Acknowledgments

Z. Wang is in part supported by NSF Award 2145346 (CA-REER). This research has been supported by computing support on the Vista GPU Cluster through the Center for Generative AI (CGAI) and the Texas Advanced Computing Center (TACC) at the University of Texas at Austin.

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

# A. Proofs

## A.1. Detailed Assumptions

**Assumption A.1. Lipschitz Hessian:** There exists $L_H > 0$ such that $\|H(W') - H(W)\| \le L_H \|W' - W\|$.

This assumption is justified by the smoothness of common loss functions (e.g., cross-entropy) and regularization terms.

**Assumption A.2. Kurdyka–Łojasiewicz (KŁ) Condition:** The loss satisfies a KŁ inequality near $W^*$, implying $\|\nabla L(W_t)\|$ decays at a power-law rate.

This assumption generalizes convexity to nonconvex.

**Assumption A.3. Spectral Gap:** For $H_t = \nabla^2 L(W_t)$ with eigenvalues in descending order, there is a uniform $\gamma > 0$ such that $\min_{1 \le i \le r, j > r} |\lambda_i(H_t) - \lambda_j(H_t)| \ge \gamma$.

This assumption is reasonable due to the dominance of task-relevant directions in the hessian spectrum, reinforced by regularization.

**Assumption A.4. Reversible Structure and Gradient Representation:** Assume the model architecture is reversible (Tian et al., 2021), ensuring no unbounded growth in norms. In particular, the gradient can be expressed as (introduced in (Zhao et al., 2024b)):

$$G_t = A_t - B_t W_t C_t$$

where $A_t, B_t, C_t$ are bounded linear mappings. Such reversibility implies uniform boundedness of $\|W_t\|$ and $\|G_t\|$ over $t$, and stable spectral properties of $H_t$.

## A.2. Proof of Theorem 2.1

To show the stabilization of the eigenspace, we start by considering the dynamics of the Hessian updates. The weights are updated using SGD as:

$$W_{t+1} = W_t - \eta \nabla L(W_t), \tag{2}$$

where $\eta$ is the learning rate.

Using a Taylor expansion, the Hessian at $W_{t+1}$ can be expressed as:

$$H_{t+1} = \nabla^2 L(W_t) + \Delta H_t, \tag{3}$$

where:

$$\Delta H_t = \nabla^2 L(W_{t+1}) - \nabla^2 L(W_t). \tag{4}$$

From the Lipschitz continuity of $\nabla^2 L(W)$, we have:

$$\|\Delta H_t\| \le L_H \|W_{t+1} - W_t\|. \tag{5}$$

Substituting $W_{t+1} - W_t = -\eta \nabla L(W_t)$, it follows that:

$$\|\Delta H_t\| \le \eta L_H \|\nabla L(W_t)\|. \tag{6}$$

From the Kurdyka-Łojasiewicz (KŁ) condition, the gradient norm satisfies:

$$\|\nabla L(W_t)\| \le \phi'(L(W_t) - L(W^*))^{-1} \tag{7}$$

Using $\phi(s) = s^{1-\theta}$, we have $\phi'(s) = (1 - \theta)s^{-\theta}$. Substituting this:

$$\|\nabla L(W_t)\| \le \frac{1}{1-\theta}(L(W_t) - L(W^*))^\theta \tag{8}$$

Let $D = (L(W_0) - L(W^*))^{1-\theta}$, representing the scale of the initial loss gap. Then:

$$L(W_t) - L(W^*) \le \frac{D}{t^{1/(1-\theta)}}. \tag{9}$$

Substituting this into the gradient bound:

$$\|\nabla L(W_t)\| \le \frac{C}{t^{\theta/(1-\theta)}}. \tag{10}$$

Thus, the bound on $\|\Delta H_t\|$ becomes:

$$\|\Delta H_t\| \le \frac{\eta L_H C}{t^{\theta/(1-\theta)}}. \tag{11}$$

Applying Weyl's inequality, the change in eigenvalues between $H_t$ and $H_{t+1}$ is bounded by:

$$|\lambda_i(H_{t+1}) - \lambda_i(H_t)| \le \|\Delta H_t\|. \tag{12}$$

Summing over $t$ bounds the cumulative eigenvalue changes:

$$|\lambda_i(H_T) - \lambda_i(H_0)| \le \sum_{t=0}^{T-1} \frac{\eta L_H C}{t^{\theta/(1-\theta)}}. \tag{13}$$

The series converges because $t^{-\theta/(1-\theta)}$ decays faster than $1/t$, ensuring the cumulative shift remains bounded, thus proving eigenvalue stabilization.

For eigenspace stabilization, the Davis-Kahan theorem provides a bound on the eigenspace shift:

$$\|U_{t+1} - U_t\| \le \frac{\|\Delta H_t\|}{\gamma}. \tag{14}$$

Substituting the bound for $\|\Delta H_t\|$:

$$\|U_{t+1} - U_t\| \le \frac{\eta L_H C}{\gamma t^{\theta/(1-\theta)}}. \tag{15}$$

Summing over $t$ bounds the cumulative eigenspace shifts:

$$\|U_T - U_0\| \leq \sum_{t=0}^{T-1} \frac{\eta L_H C}{\gamma t^{\theta/(1-\theta)}}. \tag{16}$$

The series converges for $\theta > \frac{1}{2}$, ensuring actual convergence of $U_t$. For $\theta \in \left(0, \frac{1}{2}\right]$, this analysis does not guarantee convergence, but it still provides a uniform bound on the step-to-step change in each finite interval.

### A.3. Proof of Theorem 2.2

Consider the gradient update $W_{t+1} = W_t - \eta G_t$. By Taylor's theorem, the gradient at the next step is given by:

$$\begin{aligned} G_{t+1} &= \nabla_W L(W_{t+1}) \\ &= \nabla_W L(W_t) + H_t(W_{t+1} - W_t) + R_t, \end{aligned} \tag{17}$$

where the remainder term $R_t$ is defined as:

$$\begin{aligned} R_t &= \int_0^1 (1-\tau)\nabla_W^3 L(W_t + \tau\Delta W_t)[\Delta W_t, \Delta W_t]\, d\tau, \\ \Delta W_t &= -\eta G_t. \end{aligned} \tag{18}$$

Substituting $\Delta W_t$ into the expression, we have:

$$G_{t+1} = G_t - \eta H_t G_t + R_t. \tag{19}$$

Given that $H(W)$ is Lipschitz and $\|G_t\|$ is bounded, it follows that $\|R_t\| = O(\eta^2 \|G_t\|^2)$. Thus, the update simplifies to:

$$G_{t+1} = G_t - \eta H_t G_t + O(\eta^2 \|G_t\|^2). \tag{20}$$

For each time step $t$, let $U_t = [u_1, \ldots, u_r]$ be the orthonormal eigenbasis of $H_t$ corresponding to the eigenvalues $\lambda_1, \ldots, \lambda_r$. The orthogonal complement is spanned by $V_t = [v_{r+1}, v_{r+2}, \ldots]$. The Hessian can be decomposed as:

$$H_t = U_t \Lambda_t^{(r)} U_t^\top + V_t \Lambda_t^{(-r)} V_t^\top, \tag{21}$$

where $\Lambda_t^{(r)} = \text{diag}(\lambda_1, \ldots, \lambda_r)$ and $\Lambda_t^{(-r)} = \text{diag}(\lambda_{r+1}, \ldots)$.

Decomposing the gradient $G_t$ gives:

$$G_t = U_t U_t^\top G_t + (I - U_t U_t^\top) G_t. \tag{22}$$

Applying the Hessian to the gradient, we have:

$$H_t G_t = U_t \Lambda_t^{(r)} U_t^\top G_t + V_t \Lambda_t^{(-r)} V_t^\top G_t. \tag{23}$$

Projecting the recurrence onto the non-dominant subspace results in:

$$\begin{aligned} (I - U_t U_t^\top) G_{t+1} &= (I - U_t U_t^\top) G_t \\ &\quad - \eta(I - U_t U_t^\top) H_t G_t + (I - U_t U_t^\top) R_t. \end{aligned} \tag{24}$$

Since $(I - U_t U_t^\top) H_t G_t = V_t \Lambda_t^{(-r)} V_t^\top G_t$, it follows that:

$$\begin{aligned} (I - U_t U_t^\top) G_{t+1} &= (I - U_t U_t^\top) G_t \\ &\quad - \eta \sum_{j>r} \lambda_j v_j v_j^\top G_t + (I - U_t U_t^\top) R_t. \end{aligned} \tag{25}$$

The reversible structure ensures that $\|W_t\|$, $\|G_t\|$, and $\|H_t\|$ remain uniformly bounded. This boundedness prevents pathological behavior and ensures the spectral gap $\gamma$ remains stable.

As $\{W_t\}$ is bounded and $H(\cdot)$ is Lipschitz, $\{H_t\}$ forms a bounded continuous family of matrices. Davis–Kahan perturbation theory guarantees that the eigenspaces depend continuously on $t$. The spectral gap $\gamma > 0$ ensures no mixing between the dominant and non-dominant eigenspaces, allowing $U_t$ to vary smoothly with $t$.

For $j > r$, $\lambda_j \leq \lambda_r - \gamma$. Setting $\bar{\lambda} = \lambda_r - \gamma$, we have:

$$\sum_{j>r} \lambda_j v_j v_j^\top G_t \preceq \bar{\lambda}(I - U_t U_t^\top) G_t. \tag{26}$$

Thus:

$$\begin{aligned} \|(I - U_t U_t^\top) G_{t+1}\| &\leq \|(I - U_t U_t^\top) G_t\| \\ &\quad - \eta \bar{\lambda} \|(I - U_t U_t^\top) G_t\| \\ &\quad + \|(I - U_t U_t^\top) R_t\|. \end{aligned} \tag{27}$$

Since $\|R_t\| = O(\eta^2 \|G_t\|^2)$ and $\|G_t\|$ is bounded, there exists $K > 0$ such that $\|R_t\| \leq K\eta^2$. Therefore:

$$\|(I - U_t U_t^\top) G_{t+1}\| \leq (1 - \eta\bar{\lambda})\|(I - U_t U_t^\top) G_t\| + K\eta^2. \tag{28}$$

For sufficiently small $\eta$, say $\eta < \frac{\bar{\lambda}}{2K}$, we have:

$$\|(I - U_t U_t^\top) G_{t+1}\| \leq \left(1 - \frac{\eta\bar{\lambda}}{2}\right)\|(I - U_t U_t^\top) G_t\|. \tag{29}$$

This shows a geometric contraction of the non-dominant component. As $t \to \infty$, $\|(I - U_t U_t^\top) G_t\|$ decays at least geometrically. Since $\|G_t\|$ is bounded and does not grow, the non-dominant part shrinks to zero relative to $\|G_t\|$. Hence:

$$\lim_{t \to \infty} \frac{\|(I - U_t U_t^\top) G_t\|}{\|G_t\|} = 0. \tag{30}$$

This proves that $G_t$ asymptotically aligns with the dominant eigenspace of $H_t$.

## B. Additional Experiments

### B.1. Visualization of Layer-wise Non-uniform Rank Ratio of WeLore-COMP

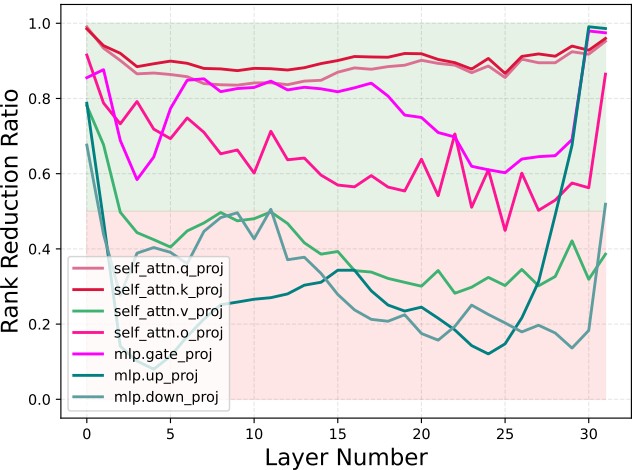

*Figure 7.* Layer-wise rank ratio of 50% compressed LLaMa-7B.

We investigated the layer-wise rank reduction ratio achieved by WeLore-COMP and found it to be highly non-uniform where some layers can be compressed significantly higher than others. In addition, note that layers from the first and last few transformer blocks are compression-friendly. Figure 7 illustrates the rank reduction ratios after 50% effective rank reduction of LLaMa-2 7B pretrained checkpoint using WeLore. Interestingly, it can be noted that self_attn.q_proj & self_attn.k_proj layers can be expressed as low-rank with > 90% compression. Moreover, the majority of layers from transformer blocks at the front and tail end are better at compression aligning with Section 2.5. The green region indicates LRCs while the red region indicates the N-LRCs components.

### B.2. Investigating further Compression Opportunity with SoTA LLM Pruning.

Recently (Yin et al., 2023) investigated the activation outlier-based non-uniform sparsity ratios for different transformer blocks within LLMs. A careful observation of their layer-wise sparsity ratio reveals that the majority of middle transformer blocks can be subjected to a higher pruning ratio which is **complementary** to WeLore low-rank reduction ratio that favours terminal blocks being low-rank friendly. We therefore ask an unexplored question: *How does LLM performance changes when we further compress only the dense N-LRCs using SoTA pruning methods?*

Figure 8 presents the increase in the perplexity of LLaMa-2 7B on the C4 dataset when we compress a dense check-point (blue) using SoTA LLM pruning methods. We compared it with further compressing dense N-LRCs of WeLore checkpoints with ERR of 10%, 30%, and 50%. Our key observations are: (i) WeLore checkpoints can further enjoy high compression with sparsification of dense N-LRCs without signification performance drop to a noticeable sparsity ratio (*e.g.,* WeLore checkpoint with ERR of 50% can be additionally sparsified using Wanda (Sun et al., 2023) with < 2 points increase in perplexity); (ii) ad-hoc sparsification of LRCs and N-LRCs (dense) suffers higher performance degradation compared to N-LRCs which demands actively exploring amalgamation of different compression techniques for LLMs to ripe maximum benefits; (iii) development of better sparsity algorithms (*e.g.,* Wanda (Sun et al., 2023), SparseGPT (Frantar & Alistarh, 2023)) clearly retain their benefits even in mixed compression settings.

## C. Background Work

**Memory-Efficient Finetuning:** Memory-efficient finetuning of LLMs aims to address the significant costs associated with their fine-tuning. This field encompasses several notable techniques. For instance, Prompt Learning Methods optimize input tokens or embedding while keeping the model's remaining parameters static (Hambardzumyan et al., 2021; Zhong et al., 2021). Layer-freezing techniques enhance training efficiency by selectively freezing certain layers (Liu et al., 2021; Brock et al., 2017; Li et al., 2024). Additionally, Adapter Methods introduce a small, update-focused auxiliary module into the model's architecture, significantly reducing the number of trainable parameters, as introduced by (Houlsby et al., 2019; Diao et al., 2022). Among them, one noteworthy technique is Low-Rank Adaptation (LoRA) (Hu et al., 2021a) and its successors (Renduchintala et al., 2023; Sheng et al., 2023; Xia et al., 2024; Zhang et al., 2023; Hayou et al., 2024; Hao et al., 2024; Liu et al., 2024), which introduces a low-rank weight adapter for each layer to reduce the memory footprint by only optimizing the adapter. These low-rank adapters can then be seamlessly merged back into the original model.

Unlike LoRA which performs proxy optimization over additional parameters while keeping the original parameters frozen, WeLore backed by an understanding of gradient dynamics suggests finetuning the original parameters of LRCs in represented in low-rank to mimic full-finetuning. Recently, (Biderman et al., 2024a) found that full finetuning is more accurate and sample-efficient than LoRA across several task categories and WeLore can be an effective alternative to achieve the benefits of full-finetuning within a limited compute and memory budget.

**Low Rank Compression:** Large Language Models (LLMs) have succeeded remarkably across various natural language processing tasks. However, the massive scale of these mod-

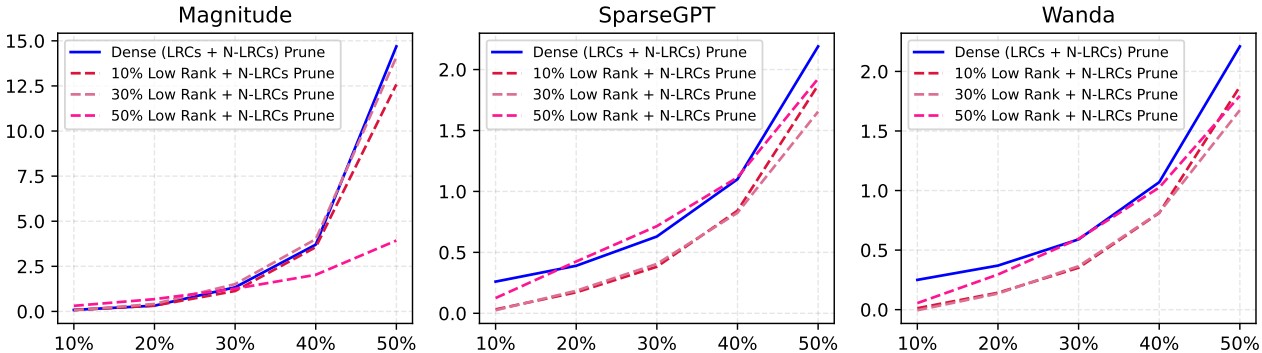

*Figure 8.* Perplexity comparison (↑) of compression of N-LRCs using SoTA LLM pruning methods for LLaMa-2 7B on C4. Here we calculated the increase in perplexity wrt. the initial perplexity of dense and low-rank compressed checkpoints with ERR of $r\%$.

els poses significant challenges in terms of storage efficiency and computational complexity. Among several techniques of LLM compression (*e.g.,*pruning, quantization, etc.), low-rank decomposition which retains only the top-k components in the low-rank space have special privilege to leverage the existing highly efficient kernels over floating point matrices. (Hsu et al., 2022a) developed a data-aware modification of SVD that incorporates approximate second-order gradient information. Similarly, (Yuan et al., 2023) proposed a data-aware decomposition method that minimizes activation error. One primary drawback of these reductions is that they uniformly reduce rank across all weight matrices. In contrast, our work experimentally validates existence of non-uniform low-rank expressiveness across different layers and should be accounted for during low-rank compression. Recently, (Zhao et al., 2023; Wang et al., 2023a) found that dynamic rank selection during pretraining can achieve comparable prediction performance as full-rank counterpart.

# D. Implementation Details

## D.1. Network Architectures:

For understanding gradient dynamics and its consequent on the weight space during pretraining, we adopt the LLaMa-130M architecture following (Lialin et al., 2023b; Zhao et al., 2024a). For our continual and downstream finetuning experiments, we adopted the pretrained checkpoint of LLaMa-2 7B and LLaMa-2 13B, and Mistral-7B from HuggingFace.

## D.2. Low Rank Compression:

For low-rank compression using WeLore for LLaMa-2 7B and 13B models, we used `torch.svd($W_l$)` to decompose a layer $l$'s weight matrix $W_l^{m\times n} = A^{m\times r}B^{r\times n}$ where r is decided by the heavy tail distribution of the singular values of $W$ as described in Section 3. If $W$ belongs to LRCs, it will be replaced with a composition of two linear layers with low-rank matrices $A$ & $B$ to improve the computational

efficiency. For baselines, we compared with commonly used uniform rank reduction (Hsu et al., 2022a; Kaushal et al., 2023a) and adopted recently proposed outlier-weighed non-uniform ratio (OWL) (Yin et al., 2023). We additionally augmented activation-guided SVD techniques (Yuan et al., 2023) with WeLore's adaptive layer-wise rank reduction ratio to understand how it can benefit them.

## D.3. Adaptive Threshold Selection

---

**Algorithm 1** Adaptive Threshold Selection Algorithm

**Input:** LLM with weights $\theta$, target reduction ratio $s_p$, current reduction ratio $s_t$, reduction tolerance $s_\delta$, threshold increment $H_i$

**Output:** A compressed model $\theta$ satisfying the target reduction ratio $s_p$, singular threshold $H$

**Initialization:** Initialize a singular threshold $H = 0$

**While *not*** $(s_p + s_\delta > s_t > s_p - s_\delta)$

 **for** each MLP layer tensor $\theta^l$ in $\theta$ **do**

  $sv^l \leftarrow$ `calculate_singular_values`$(\theta^l)$;

  $sv_n^l \leftarrow$ `normalize_singular_values`$(sv^l)$;

  $p^l \leftarrow 0$; **for** each $s$ in $sv_n^l$ **do**

   **if** $s < H$ **then**

    $p^l \leftarrow p^l + 1$;

 $P_r \leftarrow \sum_l p^l$;

 $s_t \leftarrow P_r/P_t$;

 **if** $s_p + s_\delta \geq s_t \geq s_p - s_\delta$ **then**

  **break**;

 **else**

  $H \leftarrow H + H_i$;

---

## D.4. Evaluation Tasks Settings

### D.4.1. FACTOID-BASED QUESTION ANSWERING

**Task Definition and Rationale.** Factoid-based Question Answering (Factoid-QA) (Iyyer et al., 2014), which asks precise facts about entities, is a long-standing problem in NLP.

A typical Factoid-QA task aims to search for entities or entity attributes from a knowledge graph, and it is widely used as a tool in academia, commercial search engines, and conversational assistants. Modern LLMs are trained on gigantic text corpora ingesting a large amount of world knowledge about entities and their relationships during pre-training, and have unique abilities to generate factually correct responses to user queries. In this task setting, we aim to investigate *how our WeLore-COMP based compression impacts LLMs' ability to answer natural language questions using facts, i.e., entities or attributes knowledge ingested within them during pre-training?*

**Dataset Details.** We use FreebaseQA (Jiang et al., 2019) which is a dataset for open-domain QA over the Freebase knowledge graph. The QA pairs are collected from various sources, including the TriviaQA dataset (Joshi et al., 2017) and other trivia websites (QuizBalls, QuizZone, KnowQuiz), and are matched against Freebase to generate relevant subject-predicate-object triples that were further verified by human annotators. TriviaQA dataset shows rich linguistic variation and complexity, making it a good testbed for evaluating knowledge ingested within LLMs.

D.4.2. IN-CONTEXT VARIABLE LENGTH TEXT
       SUMMARIZATION

**Task Formulation and Details.** Modern LLMs have shown astonishing success in summarizing long-context documents in both abstractive and extractive settings. However, it is **yet not explored** how FFN block skipping impacts LLMs' capability for summarization. In this task setting, we aim to investigate *how well WeLore-COMP compressed models hold onto consistency, coherence, fluency, and relevance when prompted to summarize textual information of varying length (small, medium, and large) in abstractive setting* (Jain et al., 2023). For evaluation, similar to (Zheng et al., 2023), we propose to use GPT-4 as a judge, which compares the compressed LLM generated summaries wrt. GPT-3.5 (text-davinci-003) generated summaries.

**Dataset Details and Results** We use a popular summarization dataset CNN/DailyMail (Chen et al., 2016; Jaiswal et al., 2023) for evaluation, which is an English-language dataset containing just over 300k unique news articles written by journalists at CNN and DailyMail. We created 3 subset categories {small (≤470 words), medium (≥470 and ≤ 790 words), and large (≥ 790 words)} of stories, each with 100 articles reflecting word distribution of CNN/Daily-Mail to minimize OpenAI API costs.

**D.5. Multi-turn Conversation and Instruction Following**

**Task Formulation and Rationale.** In this task setting, we investigate *how WeLore-COMP impacts the LLMs' ability to answer open-ended questions and evaluate their multi-*

*turn conversational and instruction-following ability – two critical elements for human preference*. Evaluating AI chatbots is a challenging task, as it requires examining language understanding, reasoning, and context awareness. To compare the performance of compressed LLMs' responses, we closely follow the prompt design setting in MT-Bench (Zheng et al., 2023) using GPT-4 as a judge. We prompt GPT-4 to rate the answers generated by compressed LLMs wrt. GPT-3.5 (text-davinci-003) model based on varying metrics (*e.g.*, correctness, helpfulness, logic, accuracy, *etc.*) on a scale of $[0-10]$ with detailed explanations.

**Dataset Details.** We rely on the 80 high-quality multi-turn questions identified in MT-Bench (Zheng et al., 2023). This setting covers common-use human-centric interaction with LLMs, and focuses on challenging questions to differentiate models. We used 8 common categories of user prompts to guide the prompt construction to interact with compressed LLMs: writing, roleplay, extraction, reasoning, math, coding, *etc*. For each category, we adopted manually designed 10 multi-turn questions from MT-Bench to evaluate our compressed models.

**D.6. Continual and Downstream Finetuning:**

For continual finetuning settings, we finetune the WeLore compressed LLaMa-2 7B and 13B models at different compression ratios using C4 dataset. The C4 dataset is a massive collection of Common Crawl's web crawl corpus, meticulously filtered and cleaned to ensure high-quality language modeling and training. For downstream task finetuning of compressed models, we consider a good mixture of tasks from commonsense reasoning and math reasoning, namely `CommonsenseQA`, `BoolQ`, `CoinFlip`, `SVAMP`, `BigBench`, `StrategyQA`. For comparison, we have used two baselines: (i) `LoRA`: LoRA (Hu et al., 2021a) introduces low-rank adaptors for training the models, $W = W_0 + UV$, where $W_0$ is the pretrained weights, which are frozen during training. In our setting, we associate $U$ and $V$ with all the components of the LRC and N-LRC of the compressed model and fine-tune them while keeping $W_0$ frozen. (ii) `GaLore` (Zhao et al., 2024a): GaLore projects the gradient into low-rank format and updates the optimizer states and projects it back for updating weights. In this setting, we perform finetuning of both LRCs and N-LRCs (full-finetuning) with projected low-rank gradients. Note that all our finetuning experiments start from the same checkpoint and hyperparameter settings for fair comparison.

# E. Inference GPU Memory Statistics of WeLore-COMP

In this section, we investigate the memory requirement for inference with WeLore compressed models. Table 7 how WeLore allows reducing the memory requirement to load

| Reduction | Total Params | Model Memory | `seqlen = 512` | `seqlen = 1024` | `seqlen = 2048` | `seqlen = 4096` |
|---|---|---|---|---|---|---|
| 0% | 6738.42M | 13,579 MB | 14,467 MB | 15,145 MB | 17,193 MB | 24,519 MB |
| 30% | 5794.25M | 11,993 MB | 12,565 MB | 12,923 MB | 14,549 MB | 20,853 MB |
| 50% | 4543.67M | 9,501 MB | 10,125 MB | 10,433 MB | 12,049 MB | 18,377 MB |
| 70% | 3072.84M | 6,657 MB | 7,285 MB | 7,625 MB | 9,233 MB | 15,549 MB |

*Table 7.* Empirical estimate of Inference GPU Memory Requirement (measured with GPUtil library) of LLaMa-2 7B compressed with WeLore with varying context sequence length.

the model parameters by substituting the full-rank weight matrices in their low-rank format. Given a consumer-grade GPU like GeForce RTX 4090, WeLore can facilitate inference with 4K context length where the original model will flag an OOM error.

