# OpenReview forum: "From Low Rank Gradient Subspace Stabilization to Low-Rank Weights: Observations, Theories, and Applications"
_ICML.cc/2025/Conference — ICML 2025 poster_

### Official Review · Reviewer_bRJ5 · 2025-03-07

**Overall Recommendation:** 4

**Summary:**

The authors deal with the task of studying low-rank compressions within LLM approaches studying the low-rankness of the weight matrices of the LLM.

**Claims And Evidence:**

The authors discuss the low-rankness of the matrices within the LLM model. The illustrate this numerically, which is nicely shown in Figure 1.

**Essential References Not Discussed:**

I did not miss any key references.

**Experimental Designs Or Analyses:**

The tests seemed appropriate to me.

**Methods And Evaluation Criteria:**

Yes this is appropriately chosen.

**Other Comments Or Suggestions:**

Threshold is misspelled throughout the figures

**Other Strengths And Weaknesses:**

**Strength**
**Weakness** The authors in Section 2.4 discuss sparsity but it is not clear to me what the mean here. If they refer to a low-rank representation this is quite something else than sparsity. If you mean low-rankness please be specific if you mean sparsity what is inducing this sparsity, i.e. in the coefficients, in the matrices? Again when talking about Q/K projections it is unclear to me whether sparsity is confused with low-rankness. Same comment regarding the MLP Gate Projections.

**Questions For Authors:**

Would it be possible to rescale all subfigures in Figure 1 to have the largest value at 1. This makes the rank decay more comparable. Is equation (1) really meant to be the sum of sums? Isnt the length of $S_{W_l}$ not the same everywhere? Then the numerator sums up the singular values, right? How is the activate SVD defined? This is not explained but heavily used.

**Relation To Broader Scientific Literature:**

The findings are presented in comparison to other low-rank techniques for LLMs and this seems to the best of my knowledge appropriate.

**Theoretical Claims:**

The authors prove something about the properties of the Hessian during training where the full proof is given in the appendix and a sketch in the main paper. Also the authors prove some eigenspace alignment.

---

> ### Author Rebuttal · Authors · 2025-03-31
>
> We would like to thank you for your time to review our work. Next, we will try to address the weakness pointed by you one-by-one as follows:
>
> **1. Discussion of Sparsity in Section 2.4 and typoes:** Thank you for identifying this mistake. In Section 2.4, we mean low-rank representation and we promise to correct this in our camera-ready version of our submission. We will also correct additional typoes including threshold.
>
> **2. Rescaling Figure 1:** We would like to highlight that we intentionally have not rescaled all subfigures in Figure 1 to effectively capture/illustrate the variation in eigenvalues across different component types (MLP vs attention) of the model.
>
> **3. Clarification regarding Equation 1:** $S_{W_l}$ is an array of singular values of weight matrix $W$ of layer $l$. $(S_{W_l} < k)$ will be a $0/1$ array indicating which singular values are less  than k which will be truncated or compressed. $sum(S_{W_l} < k)$ is the sum of 1’s in the array illustrating the reduction ratio of layer l. Summing across all the layers (i.e., $\sum_l sum(S_{W_l} < k)$) will give the total rank reduction in the entire model. $len(S_{W_l})$ is the indicator of the number of singular values in the weight matrix of layer $l$ and it may vary depending upon the model architecture choices.

---

> > ### Comment · Reviewer_bRJ5 · 2025-04-03
> >
> > Thank you for the comment. My main concern regarding the scaling was that it seems sensible to scale all singular values by the largest one as \frac{\sigma_i}{\sigma_{max}} in order to define a cut-off tolerance across the different matrices that makes the truncation comparable.

---

> > > ### Author Response · Authors · 2025-04-03
> > >
> > > We thank you for further clarifying your concern regarding the scaling and it is indeed a good suggestion. We will surely address this in our revised draft.
> > >
> > > If you feel that your remaining concerns have been resolved, we would greatly appreciate it if you could consider raising the score.
> > >
> > > Thank you once again for your time. We hope you have a wonderful day!
> > >
> > > Best wishes,
> > > The Authors

---

### Official Review · Reviewer_A6uE · 2025-03-13

**Overall Recommendation:** 3

**Summary:**

This paper proposes that repeated gradient alignment on the leading Hessian directions gradually drives large transformer models toward low-rank weight configurations. The authors formalize this tendency as a “rank collapse” that can be exploited both for compressing pretrained networks and for selectively finetuning only the most significant parameter subspaces.
Their method, WeLore, implements a global thresholding rule to prune weaker singular values layer by layer. Empirically, they show that WeLore not only yields strong compression with minor performance impact, but also supports a novel partial finetuning routine that updates only the high-importance (low-rank) components.

**Claims And Evidence:**

The key claim is that top Hessian eigenvectors dominate gradient trajectories in large transformers, effectively concentrating the updates within a small subspace. Data from LLaMA-family checkpoints supports this assertion, revealing that the ratio of retained singular values decreases in deeper layers without degrading perplexity or downstream accuracy. The authors also show that once low-rank behavior emerges, subsequent training maintains it or even accentuates it, implying a persistent structural bias that WeLore can exploit.

**Essential References Not Discussed:**

N/A

**Experimental Designs Or Analyses:**

In addition to testing different compression ratios on LLaMA-7B, 13B, and Mistral-7B, the authors compare WeLore’s adaptive thresholding with uniform SVD, outlier-based factorizations, and competing methods for parameter-efficient finetuning. They track how reduced-rank layers retain strong performance across open-domain QA and abstractive summarization.
Although the evaluations are thorough within the LLaMA family, no experiments on non-decoder architectures (e.g., T5) or specialized tasks (like code completion) appear. This leaves questions about broader applicability.

**Methods And Evaluation Criteria:**

WeLore applies a global singular-value threshold to each layer’s weight matrices, creating an adaptive truncation that balances compression and fidelity. The authors assess the resulting models on perplexity (using the C4 dataset) and multiple downstream tasks (QA, summarization, conversation).
They also measure speedups from partial finetuning, focusing on the memory and throughput gains by only updating the “highly ranked” fraction of parameters.

**Other Comments Or Suggestions:**

It would be valuable to investigate whether these low-rank patterns hold in tasks that deviate strongly from the original training domain, such as specialized code-generation or multimodal inputs.

**Other Strengths And Weaknesses:**

A notable strength is the demonstration that partial finetuning on just the “most relevant” components can match or exceed full finetuning in some tasks, potentially reducing hardware requirements in practical scenarios. Another strength is how their single threshold mechanism adjusts itself to each layer’s singular values.

However, a potential weakness is that the approach has only been tested on decoder-only or Mistral-like models, so it is unclear if encoder-decoder architectures with cross-attention layers would exhibit comparable rank collapse. Also, the reliance on repeated SVD across many layers might be resource-intensive, and the discussion of approximate factorization is brief.

**Questions For Authors:**

Please see several questions in my above comments. Two additional questions:
1. Can deeper cross-attention layers or encoder-decoder styles disrupt the rank collapse pattern observed in decoder-only frameworks, and if so, how might that influence WeLore’s thresholding strategy?
2. Does “freezing” the bulk of weights ever lead to underfitting for tasks that place heavy demands on representational flexibility, or does the initial training’s rank deficiency remain adequate for adaptation?

**Relation To Broader Scientific Literature:**

This work situates itself at the pretty unique intersection of Hessian-based analyses of network optimization and low-rank compression strategies.
Unlike prior studies that introduce side modules (e.g., LoRA) or uniform rank constraints, WeLore leverages a single threshold that reflects global curvature trends. It also resonates with recent theoretical models linking gradient flow to emergent structure in overparameterized networks, though the paper’s emphasis on multi-layer transformers marks a specialized application.

**Theoretical Claims:**

The analysis builds on classic Lipschitz continuity arguments and eigenvector perturbation bounds, positing that second-order curvature narrows the gradient search space over training iterations.
By showing that the Hessian’s top eigenvalues shift slowly, the authors infer that meaningful weight updates stay within a lower-dimensional manifold, effectively turning the model’s parameters into a near-rank-deficient collection.
While they provide partial proofs for these curvature-based claims, the full set of constants and domain-specific constraints (like attention mechanisms) remain only informally addressed.

---

> ### Author Rebuttal · Authors · 2025-03-31
>
> We would first like to thank you for the time to review our work. We would now address your weakness point by point as follows:
>
> **1. Computational overhead of SVD on large matrices:** Thank you for raising this point. We would like to highlight that WeLore-COMP is a *one-shot data-agnostic* compression technique. WeLore-COMP requires **one-time SVD estimation** of all weight matrices of the pretrained checkpoint which can be saved and various compression ratios (e.g., 10%, 20%, 30%, etc.) can be achieved by a simple linear search of threshold k (Algorithm 1) without any need for re-estimation of SVD.
>
> 1.  For empirical estimates of **LLaMa2-7B** using a Nvidia A6000 RTX GPU, a 4096 * 4096 matrix SVD decomposition takes ~2.6895 seconds to complete. Given 7 weight matrices for each transformer layer of approximately similar dimensions in LLaMa2, SVD time cost for decomposition of a transformer layer will be ~19 seconds. With 32 layers, the total time taken will be approximately ~10 minutes. *Note that SVD estimation of each layer can be executed independently and parallel*. Therefore, with 8 GPUs, we can reduce the time 10/8 = ~1.25 minutes in total.
>
> 2.  For empirical estimates of **LLaMa2-70B** using a Nvidia A6000 RTX GPU, a 8192 * 8192 matrix SVD decomposition takes ~2.799 seconds to complete. Given 7 weight matrices for each transformer layer of approximately similar dimensions in LLaMa2, SVD time cost for decomposition of a transformer layer will be ~20 seconds. With 80 layers, the total time taken will be approximately ~26 minutes. *Note that SVD estimation of each layer can be executed independently and parallel*. Therefore, with 8 GPUs, we can reduce the time 26/8 = ~3.25 minutes in total.
>
> We promise to include additional computational cost discussion on SVD in our final draft.
>
> **2. Low-rank patterns and Datasets:** We would like to clarify that low-rank patterns emerging in model checkpoints are subjected to their pre-training. WeLore exploits these existing patterns for data-agnostic compression and PEFT. Experimentally through a wide-range of tasks in our experiments ranging from commonsense tasks to open-ended tasks (MT-Bench with code-centric questions - Table 2), we found that WeLore consistently perform well for the existing low-rank patterns in pre-trained checkpoints.
>
> **3. Freezing Weights and Underfitting:** Thank you for raising this point. Across all the experiments with WeLore-PEFT with different compression ratios, we didn’t observe the underfitting issue and WeLore-PEFT can closely mimic the training trajectory of full-finetuning. For example, as it can be seen from Table 6, with 30% compression, which approximately freeze 70% weights of the model during PEFT, WeLore outperform full-finetuning across several tasks which indicate that task adaptations can be adopted by only a few components in the model.
>
> **4. Extension for encoder-decoder architecture:** Thank you for this suggestion. Decoder-only architectures like LLaMa and Mistral family are most popular and scaled up architecture, that’s why we choose them as our experimental focus. We would like to highlight that while deeper cross-attention layers or an encoder-decoder structure might exhibit differences in rank collapse behavior compared to decoder-only models, these differences do not fundamentally invalidate the mechanism underlying WeLore’s compression or PEFT strategy. The low-rank collapse is driven by the stabilization of the gradient subspace and the emergence of a clear Hessian spectral gap—a phenomenon that is intrinsic to the training dynamics rather than being solely an artifact of a particular architecture. Even if cross-attention layers show a less pronounced rank collapse (due to their role in integrating encoder context, which can diffuse gradient signals), WeLore’s design is inherently adaptive. In other words:
>
> • Even if some layers (such as deep cross-attention ones) deviate from the typical rank collapse pattern, the thresholding in WeLore is computed on a per-layer basis, automatically preserving higher ranks where needed and applying more aggressive compression where the low-rank structure is clear.
>
> • The core mechanism—gradient subspace alignment with dominant Hessian directions—remains relevant across different architectures. Thus, while encoder-decoder models might introduce variations, they do not “disrupt” the phenomenon in a way that would render adaptive thresholding ineffective.
>
> Due to time limit of rebuttal, we couldn't complete our experiments but we are working towards it and we will include in our final version.

---

### Official Review · Reviewer_XxZn · 2025-03-14

**Overall Recommendation:** 3

**Summary:**

This paper studies the emergence of low-rank structures in Large Language Models (LLMs) through gradient subspace stabilization, revealing that as training progresses, gradients increasingly align with dominant Hessian eigenspaces, driving weight matrices toward low-rank factorization. The authors support this phenomenon with theoretical analysis—incorporating Hessian smoothness, KL conditions, and reversibility assumptions—and empirical evidence from LLaMA-based models (7B, 13B) and Mistral-7B. Building on these insights, they propose WeLore-COMP, a data-agnostic compression method that applies an adaptive global threshold on singular values to selectively compress "low-rank components" (LRCs) while preserving "non-low-rank components" (N-LRCs), and WeLore-PEFT, a parameter-efficient fine-tuning approach that optimizes only LRCs while freezing N-LRCs. Empirically, WeLore-PEFT matches or surpasses full fine-tuning performance while significantly reducing memory and computational costs, demonstrating the effectiveness of leveraging LLMs' intrinsic low-rank structure for efficient model compression and adaptation.

## update after rebuttal

Thanks for the response. I will keep my score.

**Claims And Evidence:**

Yes

**Essential References Not Discussed:**

N/A

**Experimental Designs Or Analyses:**

Yes - All experiments.

**Methods And Evaluation Criteria:**

Yes

**Other Comments Or Suggestions:**

N/A

**Other Strengths And Weaknesses:**

## Strengths:

The paper provides a novel theoretical explanation by linking the emergence of low-rank structures to Hessian eigenspace alignment, offering a more principled understanding beyond heuristic-based approaches. The proposed adaptive rank thresholding achieves a better balance between compression and accuracy compared to uniform baseline methods. Extensive experiments across different model sizes, fine-tuning settings, and tasks demonstrate the robustness of the approach. Additionally, WeLore-PEFT has practical significance, as it can be directly integrated into training pipelines, reducing memory and compute costs while maintaining strong performance.

## Weaknesses:

1. The study focuses primarily on LLaMA- and Mistral-based models, limiting its generalizability to other architectures such as GPT-style models.
2. The computational overhead of performing singular value decomposition (SVD) on large matrices is not thoroughly analyzed, particularly for very large models (13B+ parameters), where approximate SVD methods may be necessary.
3. The theoretical framework relies on a reversibility assumption that may not always hold in standard Transformer layers, and the practical implications of this assumption are not fully discussed.
4. Some layers are classified as non-low-rank components (N-LRCs) with minimal rank collapse, but it remains unclear whether these could still benefit from advanced compression techniques or partial rank factorization.

**Questions For Authors:**

Please see weaknesses.

**Relation To Broader Scientific Literature:**

This paper extends prior works on:
1. LLM compression (e.g., SVD-based approaches, low-rank factorization, pruning, quantization).
2. PEFT (LoRA, QLoRA, and other adapter-based methods).
3. Hessian/gradient-based analyses linking spectral properties to optimization geometry.

**Theoretical Claims:**

The main theoretical insights (Theorems 2.1 and 2.2) revolve around bounding changes in Hessian eigenvalues/eigenvectors over training steps under certain smoothness and KL assumptions. They demonstrate that: (1) The top-r eigenvectors converge quickly in orientation (Davis–Kahan, implying small angles); (2) The gradient eventually resides in the span of these dominant eigenvectors, pushing the solution toward a lower-rank subspace.
While the authors do not present a fully rigorous line-by-line proof for every constant, they cite standard results (Lipschitz continuity, KL-based gradient decay) and supply sketches in Appendix A. This level of detail is acceptable for a conference paper.

---

> ### Author Rebuttal · Authors · 2025-03-31
>
> We would first like to thank you for the time to review our work. We would first like to thank you for finding our work to provide extensive experiments to establish robustness and have practical significance. We would now address your weakness point by point as follows:
>
> **1. Computational overhead of SVD on large matrices:** Thank you for raising this point. We would like to highlight that WeLore-COMP is a *one-shot data-agnostic* compression technique. WeLore-COMP requires **one-time SVD estimation** of all weight matrices of the pretrained checkpoint which can be saved and various compression ratios (e.g., 10%, 20%, 30%, etc.) can be achieved by a simple linear search of threshold k (Algorithm 1) without any need for re-estimation of SVD.
>
> 1.  For empirical estimates of **LLaMa2-7B** using a Nvidia A6000 RTX GPU, a 4096 * 4096 matrix SVD decomposition takes ~2.6895 seconds to complete. Given 7 weight matrices for each transformer layer of approximately similar dimensions in LLaMa2, SVD time cost for decomposition of a transformer layer will be ~19 seconds. With 32 layers, the total time taken will be approximately ~10 minutes. *Note that SVD estimation of each layer can be executed independently and parallel*. Therefore, with 8 GPUs, we can reduce the time 10/8 = ~1.25 minutes in total.
>
> 2.  For empirical estimates of **LLaMa2-70B** using a Nvidia A6000 RTX GPU, a 8192 * 8192 matrix SVD decomposition takes ~2.799 seconds to complete. Given 7 weight matrices for each transformer layer of approximately similar dimensions in LLaMa2, SVD time cost for decomposition of a transformer layer will be ~20 seconds. With 80 layers, the total time taken will be approximately ~26 minutes. *Note that SVD estimation of each layer can be executed independently and parallel*. Therefore, with 8 GPUs, we can reduce the time 26/8 = ~3.25 minutes in total.
>
> We promise to include additional computational cost discussion on SVD in our final draft.
>
> **2. Generalizability to other architectures such as GPT-style models:** We would like to have additional clarification on this weakness.  Our experiments across various tasks,compression ratios on LLaMa & Mistral illustrate that WeLore generalizes to GPT-style decoder models. Decoder architectures are most popular and scaled up architecture, that’s why they are our experimental focus.
>
>
> **3. Theoretical framework relies on a reversibility assumption:** The current proof employs the reversible network structure primarily as a sufficient condition to ensure the boundedness and stability of both gradients and Hessians. In particular, the reversibility assumption is invoked to prevent unbounded growth of norms and to guarantee stable spectral properties of the Hessian sequence $\{H_t\}$. Notably, if boundedness and stability can be established through alternative means, strict reversibility may not be necessary.
>
> From a purely mathematical standpoint, the key requirements of the proof are as follows:
> 1. **Boundedness of $||G||$, $||W_t||$, and $||H_t||$:**
>    It is imperative to ensure that the norms of gradients, weights, and Hessians do not diverge. When these quantities remain bounded, the spectral gap property and the Lipschitz continuity of the Hessian ensure stable eigen-decomposition and validate the application of Davis–Kahan perturbation theory.
> 2. **Stable Spectral Gap:**
>    For $H_t=\nabla^2 L(W_t)$ with eigenvalues in descending order, there is a uniform $\gamma>0$ such that $\min_{1 \le i \le r,\, j>r}|\lambda_i(H_t)-\lambda_j(H_t)|\ge \gamma$.
>
> It is important to emphasize that these conditions do not inherently require reversibility. Many standard optimization settings—such as strongly convex problems or well-conditioned neural architectures—employ mechanisms that naturally ensure boundedness and prevent degenerate Hessians. Consequently, if uniform boundedness of parameters, gradients, and Hessians can be secured by other means (for example, weight regularization, strong convexity near minima, or alternative architectural constraints), then the strict assumption of reversibility is not essential.
>
> **4. Benefit from other compression techniques on N-LRCs layers:** Thank you for raising this point and we agree that exploration of mixed compression strategy can be highly interesting with WeLore. We have indeed conducted experiments to study WeLore in conjunction with pruning. A careful observation of popular *non-uniform layerwise pruning algorithms (https://arxiv.org/pdf/2310.05175) reveals that the majority of middle transformer blocks can be subjected to a higher pruning ratio* which is **complementary** to *WeLore low-rank reduction ratio that favours terminal blocks being low-rank friendly* (Appendix B.1). Our experimental results in Appendix B.2 reveals that **WeLore can be used in conjunction with SoTA pruning techniques** like SparseGPT, Wanda etc. due to existing orthogonal properties in mixed compression settings with minimal performance degradations.

---

### Official Review · Reviewer_pkHm · 2025-03-14

**Overall Recommendation:** 3

**Summary:**

This paper investigates the low-rank property of LLM weights. The authors identify that low-rank properties vary systematically across components (q/k/v/o/mlp1/mlp2/out) and network depth. Based on this observation, they develop: 1) WeLore-COMP for non-uniform compression across different layers, and 2) WeLore-PEFT for selectively fine-tuning only components with good low-rank properties.

**Claims And Evidence:**

Yes.

1. Novel theoretical analysis. The gradient subspace analysis provides valuable insights into why low-rank structures emerge in LLMs, connecting optimization dynamics to model structure.

2. Non-uniform approach. The recognition that different components have inherently different low-rank properties is insightful and leads to significant performance gains over uniform compression methods.

**Essential References Not Discussed:**

NO

**Experimental Designs Or Analyses:**

Yes, this paper has a comprehensive evaluation. The work includes extensive experiments across different models (LLaMA-2 7B/13B, Mistral-7B), compression ratios (10-70%), and downstream tasks.

**Methods And Evaluation Criteria:**

Yes.

Unified framework. Unlike prior work focusing separately on compression or fine-tuning, WeLore provides a cohesive approach addressing both challenges simultaneously.

**Other Comments Or Suggestions:**

NA

**Other Strengths And Weaknesses:**

weaknesses:

1. Figure issues and visualization clarity. Several figure problems: (1) Row 1, column 4 in Figure 2 is mislabeled as "k_proj" instead of "q_proj"; (2) Figure 1 lacks logical organization for comparing eigenvalue gaps among MLP components, attention components, and different layer depths; (3) Almost all the texts are missing in Figures 3-6, so it is impossible to interpret their results properly.

2. Hyperparameter sensitivity. The threshold $k$ used for determining rank reduction lacks rigorous justification. The paper doesn't analyze how sensitive the results are to this parameter or provide a principled method for selecting optimal values across different models and tasks.

3. Subjective LRC/N-LRC definition. The distinction between Low-Rank Components (LRCs) and Non-Low-Rank Components (N-LRCs) relies on empirical thresholds and visual interpretation of "heavy-tail" distributions (In figures 1-2). A quantitative metric would be better.
Limited comparison with quantization methods. The method lacks thorough comparison with compression methods (quantization, pruning).

**Questions For Authors:**

NA

**Relation To Broader Scientific Literature:**

Yes

**Theoretical Claims:**

Not yet.

---

> ### Author Rebuttal · Authors · 2025-03-31
>
> We would first like to thank you for the time to review our work. We greatly appreciate that you have found our theoretical analysis novel and experimental section comprehensive while identifying the unique proposition of WeLore as a cohesive method to handle compression and fine-tuning in a unified way. We would now like to address the weaknesses pointed by you one by one as follows:
>
> **1. Issues with Figure and Visualization Clarity:** We appreciate your concerns regarding about some mislabel and missing texts in our figures in the submitted draft. We promise you that we will address all these issues in the camera-ready version of our submission.
>
> **2. Hyperparameter sensitivity (k):** Thank you for raising this point. We would like to highlight that our hyperparameter k is uniquely determined/searched based on the effective rank reduction ratio (ERR) required. We use a linear search with np.linspace(0, 1, 0.005) using the algorithm described in Appendix D3 (Algorithm 1) to preserve normalized singular values greater than the threshold k so that we can achieve the ERR. In our experiments,  we have found that the precision 0.005 is sufficient to select optimal threshold K for a given ERR without notable variation in results. We will provide this additional clarification in the final draft of our submission. For additional information, we have included the pre-estimated singular value thresholds (k) for LLaMa-2 7B and 13B as follows:
>
>
> |Model| 10% | 20% | 30% |  40% | 50% | 60% | 70%|
> | ------------- |:-------------:|:-------------:|:-------------:|:-------------:|-------------:|-------------:|-------------:|
> |LLaMa-2 7B| 0.065 | 0.085|  0.115|  0.145 | 0.175 | 0.215 | 0.260|
> |LLaMa-2 13B| 0.065  | 0.085 |  0.115 |  0.140 |  0.180 |  0.225 |  0.270 |
>
> **3. WeLore and other compression techniques:** We appreciate your interest about WeLore comparison with other compression techniques like pruning and quantization. WeLore is a low-rank compression method with its own unique benefit of hardware-friendly acceleration and it is unclear what compression ratio of low-rank compression will equate to what compression ratio of pruning and quantization. However, we have indeed conducted experiments to study WeLore in conjunction with pruning. A careful observation of popular *non-uniform layerwise pruning algorithms (https://arxiv.org/pdf/2310.05175) reveals that the majority of middle transformer blocks can be subjected to a higher pruning ratio* which is **complementary** to *WeLore low-rank reduction ratio that favours terminal blocks being low-rank friendly* (Appendix B.1). Our experimental results in Appendix B.2 reveals that WeLore can be used in conjunction with SoTA pruning techniques like SparseGPT, Wanda etc. due to existing orthogonal properties in mixed compression settings with minimal performance degradations.

---

### Decision · Program_Chairs · 2025-05-01

**Decision:**

Accept (poster)

**Comment:**

This paper introduces a unified framework for memory-efficient fine-tuning and compression of LLMs, grounded in a theoretical analysis of gradient subspace stabilization and low-rank weight matrix properties.
This submission received strong support from multiple reviewers:
- Reviewer pkHm highlighted the originality of the theoretical analysis and the unified design of WeLore
- Reviewer XxZn appreciated the theoretical grounding, strong empirical evaluation, and practicality of the method.
- Reviewer A6uE commended the demonstration that partial fine-tuning can outperform full fine-tuning under significant compression.
- Reviewer bRJ5 acknowledged the soundness of the theoretical analysis.

Reviewers acknowledged and appreciated the authors’ responses.  Before the rebuttal, Reviewers raised concerns regarding figure clarity and typographical issues, the empirical justification and sensitivity of the threshold hyperparameter, the scalability of SVD in large models, the limited exploration beyond decoder-only architectures. These issues were comprehensively addressed in the rebuttal with clear clarification, additional empirical details, and plans for revisions.

Therefore, AC recommends acceptance of the paper.